# Potassium fertilization combined with crop straw incorporation alters soil potassium fractions and availability in northwest China: An incubation study

Xiushuang Li[1], Yafei Li[1], Tianqi Wu[1], Chunyan Qu[1], Peng Ning[1,2], Jianglan Shi[1,2]*, Xiaohong Tian[1,2]

1 College of Natural Resources and Environment, Northwest A&F University/ Key Laboratory of Plant Nutrition and the Agri-environment in Northwest China, Ministry of Agriculture and Rural Affairs, Yangling, Shaanxi, China, 2 Scientific Laboratory of Heyang Agricultural Environment and Farmland Cultivation, Ministry of Agriculture and Rural Affairs, Heyang, Shaanxi, China

* shijl81@nwsuaf.edu.cn

**Data Availability Statement:** All relevant data are within the paper.

**Funding:** This work was financed by the Natural Sciences Fundamental Research Program of

## Abstract

Potassium (K) input is essential for the improvement of soil fertility in agricultural systems. However, organic amendment may differ from mineral K fertilization with respect to modifying the soil K transformation among different fractions, affecting soil K availability. We conducted a 60-day lab incubation experiment to evaluate the response of soil K dynamics and availability in various fractions with a view to simulating crop residue return and chemical K fertilization in an Anthrosol of northwest China. The tested soil was divided into two main groups, no K fertilization (K0) and K fertilization (K1), each of which was subjected to four straw addition regimes: no straw addition (Control), wheat straw addition (WS), maize straw addition (MS), and both wheat straw and maize straw addition (WS+MS). Soil K levels in the available (AK) and non-exchangeable (NEK) fractions were both significantly increased after K addition, following the order of K>WS>MS. Fertilizer K was the most efficient K source, demonstrating a 72.9% efficiency in increasing soil AK, while wheat and maize straw exhibited efficiencies of 47.1% and 39.3%, respectively. Furthermore, K fertilization and wheat and maize straw addition increased the soil AK in a cumulative manner when used in combination. The mobility factor ($M_F$) and reduced partition index ($I_R$) of soil K were used to quantitate the comprehensive soil K mobility and stability, respectively. Positive relationships were observed between the $M_F$ and all relatively available fractions of soil K, whereas the $I_R$ value of soil K correlated negatively with both $M_F$ and all available fractions of soil K. In conclusion, straw amendment could be inferior to mineral K fertilization in improving soil K availability when they were almost equal in the net K input. Crop straw return coupled with K fertilization can be a promising strategy for improving both soil K availability and cycling in soil–plant systems.

Shaanxi (2017JM4029), the Key R&D Program of Shaanxi (2019ZDLNY01-05-01), and the National Key R&D Program (2016YFD0200308), China. Their recipients were Prof. Shi Jianglan and Tian Xiaohong, who gave us some important suggestion about the conception, design, and perforation of the experiment, and the writing of the manuscript.

**Competing interests:** The authors have declared that no competing interests exist.

## Introduction

Potassium (K) is an essential macronutrient for plant growth, a large quantity of which is present in the soil within secondary clay minerals [1,2]. In the agricultural ecosystem, K plays a key nutritional role in determining crop yield [3–5]. Over recent decades, intensive agriculture has substantially increased crop production in both developed and developing countries [6–8], resulting in considerable depletion of soil K through crop removal. Soil K deficiency, especially in the fractions available to plants, is currently a worldwide problem [9–11].

In China, soil K deficiency is closely correlated with the excessive application of nitrogen (N) and phosphorus (P) [12,13]. In intensive agricultural systems, fertilizer recommendations and the subsequent application of N and P have increased annually over recent decades, whereas the application of K has not been increased accordingly [14]. Farmers are not aware of the economic benefit of K fertilizer application since it is more expensive than N and P fertilizer and does not increase crop yield as quickly [10,15]. However, results of soil fertility tests have shown that soil K content is declining nationwide, especially in originally infertile and intensive agricultural soils [5,10,12]; therefore, management aimed at mitigating the negative K budget is urgently required.

The return of crop residue is crucial for maintaining soil quality and increasing agricultural productivity since it is rich in soil organic matter and mineral nutrients [16–20]. Particularly in cereal crops such as maize and wheat, a large amount of K is present in crop straws with extremely low harvest indexes (12–18%). Moreover, the K content of plants is not linked to organic compounds and can therefore be easily released and available after straw return [21–22]. It has been well documented that both crop yield and soil K availability can be improved by long-term straw return [11,23,24]. In China, crop straw return is widely practiced in agricultural production [25,26]; however, relying only on internal circulation of the soil–plant system is not sufficient to relieve the soil K deficit. Crop straw return should be combined with K fertilization to not only offset soil K deficit, but also improve K cycling.

The fraction in which soil K resides determines its bioavailability [27,28]. Generally, soil K exists in water-soluble (WSK), exchangeable (EK), non-exchangeable (NEK), and structural (SK) fractions, among which WSK and EK are easily released into the soil and readily available to plants [2,27]. Soil EK is the dominant form of available K (AK), which is electrostatically retained on the outer surface of clay minerals and organic substances [29–31]. According to previous studies, soil EK can be further separated into non-specific adsorptive K (NSAK) and specific adsorptive K (SAK). Soil SAK is adsorbed around the edges and wedge zones of micaceous clay minerals and is more intensively held by soil minerals, with relatively lower mobility as compared with NSAK [27,32]; however, dynamic equilibrium reactions of soil K exist among different fractions [2,33]. For example, soil AK can be fixed as NEK, which reduces its bioavailability for the current crop since soil $K^+$ ions in soluble or adsorbed forms enter the adsorption sites in mineral matrixes of 2:1 type clay [2,27,31]. Therefore, the relative distribution of soil K among different fractions can be altered by variations in dynamic equilibrium, which affects soil K availability.

In turn, the redistribution of soil K among different fractions can also break the dynamic equilibrium, driving K transformation. Increased soil AK can simultaneously cause its fixation as NEK, reserving it as the major source of K for following cropping systems [2,34–36]. In some cases, the release of soil NEK to available fractions occurs when EK and WSK levels are decreased by crop removal and/or leaching [27,35]; thus, both soil AK and NEK characteristics must be considered when assessing soil K availability and exogenous K efficiency, particularly in long-term cropping systems. Moreover, soils differ in their tendency to fix applied K, with each soil having its own fixing capacity for K that is also affected by changes in soil solutions

[2,27,29]. Theoretically, soils with a relatively higher content of organic matter provide a greater number of adsorption sites for EK, protecting soil AK against fixation. However, the addition of organic carbon (C) to soil may also lead to immobilization of available nutrients by altering physical, chemical, and biological factors in the soil [20,37,38]. As a result of concomitant organic C addition, it remains uncertain whether crop straw return differs from mineral K fertilization with respect to regulating the relative distribution of AK. Further, suitable methods to assess soil K bioavailability by simultaneously considering soil AK and NEK characterization are required for long-term cultivation.

Soils in the main agricultural areas of northern China are generally developed from K-rich parent materials; hence, the soil K content is not a major limiting factor in crop production [10,11,29]. The winter wheat (*Triticum aestivum* L.)–summer maize (*Zea mays* L.) rotation system is commonly practiced in these areas and is characterized by intensive agriculture with a high input of N and P but inadequate K addition [12,14,19,39]. Despite the original abundance of K in these regions, soil K has been in deficit for decades; it is estimated that approximately 4333 kg·ha$^{-1}$ net K has been removed from the soil over the past 20 years with no K replacement [40]. Soil K is gradually limiting crop production, particularly in the major grain-producing areas of northern China [1,3,10]; thus, increasing the addition of K is urgently required in these areas to maintain crop production and improve soil fertility.

On the other hand, the use of both the mobility factor ($M_F$) and reduced partition index ($I_R$) has already been confirmed as an efficient assessment of the redistribution and bioavailability of metals in soil [41,42]. The $M_F$, which is dictated by the metal content of the soluble and exchangeable fractions, can be used to describe the metal mobility in soil. In contrast, $I_R$ expresses soil metal transformation in individual fractions and explains their redistribution. Both of these indexes are promising for the appropriate quantitation of K bioavailability of multiple fractions but reports regarding their use for this purpose are rare.

Considering this background information, we hypothesized that mineral K fertilization and straw (wheat and maize) return would differentially affect the relative distribution of soil K among various fractions, thus determining soil K availability dependent on variations in the $M_F$ and $I_R$ values. An incubation experiment was performed to monitor soil K dynamics in various fractions following simulated crop straw return, K fertilization, and a combination of both treatments. The objectives were: (i) to assess the responses of soil K in different fractions to K addition, including K fertilization and straw return; (ii) to examine how K fertilization and straw return influence soil K bioavailability when considering all K fractions, based on the observed changes in the $M_F$ and $I_R$ of soil K; and (iii) to develop the optimal management practice for the improvement of soil K availability and cycling in this intensive agricultural system.

## Materials and methods

### Soil and crop straw

An incubation experiment was carried out to determine the dynamics of soil K following straw incorporation and K fertilizer application. Soil samples (0–20 cm layer) were collected from the Northwest A&F University Experimental Station (34°17'44" N, 108°04'10" E, 524.7 m above sea level), Shaanxi Province, North China. Annual winter wheat–summer maize copping rotation has long been the dominant system in this region. Summer maize (*Zea mays* L.) is grown from June to October every year, followed by winter wheat (*Triticum aestivum* L.) from October to the following June. There has been no potash fertilizer applied in this region for decades due to the inherently K-rich soil parent materials with a total K of more than 20g kg$^{-1}$. Soil was collected from a field without straw return to avoid the crop residue effect. The

soil was classified as an Earth-cumuli Orthic Anthrosol and had a pH of 8.08, SOC of 9.92 $g \cdot kg^{-1}$, total N, P, and K of 1.17, 0.81, and 22.32 $g \cdot kg^{-1}$, respectively, and an available N, P, and K of 20.26, 10.53, and 142 $mg \cdot kg^{-1}$, respectively. The major minerals in the soil were illite and montmorillonite. Soil samples were air-dried, ground to particles <2-mm in size, and divided into two subsamples. Most of the soil was used for the incubation experiment and a small proportion was used to determine the basic physicochemical properties.

The winter wheat and summer maize straw was collected after each crop harvest in 2015, dried, and ground to <1 mm for subsequent incubation and total K analysis. The K concentrations in wheat and maize straws were 1.45% and 1.25%, respectively.

## Experimental design

The tested soil was divided into two groups, no K fertilization (K0) and 200 mg $K_2O \cdot kg^{-1}$ dry soil fertilization (K1), each of which were subjected to four simulated straw addition regimes: no straw addition (Control), wheat straw addition (WS, straw was applied at 12 $g \cdot kg^{-1}$ dry soil), maize straw addition (MS, straw was applied at 12 $g \cdot kg^{-1}$ dry soil), and both wheat and maize straw addition (WS+MS, both wheat and maize straw were applied at 12 $g \cdot kg^{-1}$ dry soil). For the laboratory incubation experiment, the treatments were arranged using a completely random design and 12 replications were performed for each treatment. For the incubation, 250 g soil (dry weight) and the corresponding amount of straw and K fertilizer ($K_2SO_4$) were added to each 1-L jar (15 cm in height, 9 cm in diameter, with a perforated cover) and mixed thoroughly. A nutrient solution containing nitrogen (urea) and $P_2O_5$ (superphosphate) was mixed with distilled water and applied to each jar to achieve 70% of the water-holding capacity of the soil. The N and P were applied at 200 $mg \, kg^{-1}$ soil and 40 $mg \, kg^{-1}$ soil, respectively, in all treated soils, to supply the metabolism of soil microorganisms. All jars were incubated in the dark for 60 days (d) at 25°C. Deionized water was added weekly, as required, to maintain a constant soil moisture by weight.

## Sampling and analyses

Soil in each jar was sampled on incubation days 15, 30, 45, and 60 by removing three replicates from each treatment condition using a destructive sampling method. Soil samples were air-dried, ground, and sieved to particles <1 mm for WSK, AK, NASK, SAK, and NEK determination, and to 0.15 mm for total K determination.

Total K (TK) in the soil was digested in a nickel crucible with sodium hydroxide at 750°C [10]. Available K (AK) was extracted using 1 $mol \cdot L^{-1}$ ammonium acetate [43]. Moreover, three solvents were used to extract the other K fractions. Water-soluble K (WSK) was extracted using distilled water. Soil exchangeable K (EK) was calculated by subtracting WSK from AK. According to previous research [27,32], soil non-specific adsorptive K (NSAK) was extracted using 0.5 $mol \cdot L^{-1}$ magnesium acetate and calculated by subtracting WSK from the K extracted using 0.5 $mol \cdot L^{-1}$ magnesium acetate. Subsequently, soil specific adsorptive K (SAK) was calculated by subtracting both WSK and NSAK from AK. Non-exchangeable K (NEK or slowly available K) was extracted using the hot nitric acid extraction method and calculated by subtracting AK from the K extracted using hot nitric acid [43]. Structural K (SK) was calculated by subtracting the K extracted using hot nitric acid from TK. Plant K in crop straw was digested using the $H_2SO_4$-$H_2O_2$ method. K concentrations in all sample solutions were determined using an atomic absorption spectrophotometer (Analyst 400, PerkinElmer, U.S.).

## Calculations and statistical analysis

The efficiency ratio of exogenous K was used to assess the apparent bioavailability after K addition to soil, which is defined as the ratio of the increase in AK relative to exogenous K addition, according to Eq 1 [11,29,31]:

$$K \text{ efficiency ratio}(\%) = {}^{(AK_T - AK_C)}\big/_{K \text{ input}} \times 100 \qquad (1)$$

where $AK_T$ is the amount of available K in the soil after exogenous K addition; $AK_C$ is the amount of available K in the soil without exogenous K addition; and K input is the net amount of K added.

Further, the mobility factor ($M_F$) was used to assess the relative K mobility and bioavailability in the soil, which is defined as the ratio of the K concentration in the mobile or available fraction relative to the sum of the K concentration in all fractions, according to Eq 2 [42,44]:

$$M_F(\%) = \frac{F1 + F2 + F3}{F1 + F2 + F3 + F4 + F5} \times 100 \qquad (2)$$

where F1–F5 are the soil levels of WSK, NSAK, SAK, NEK, and SK respectively. F1–F3 are commonly considered mobile (available) fractions of soil K.

The partition index ($I_R$) of soil metal elements describes their relative binding intensity and is widely employed in research regarding the mobility of metals or microelements in soil [42,45]. Accordingly, in the present research, $I_R$ was expressed to assess the relative binding intensity of soil K based on synthesizing various fractions using Eq 3:

$$I_R = \frac{\sum_{i=1}^{k} i^2 F i}{k^2} \qquad (3)$$

where $i$ is the index number of the K fraction, progressing from WSK for the F1 fraction to SK for the F5 fraction (dependent on our extraction method, $k = 5$), $Fi$ is the K percentage content of the considered K in the $i$th fraction. Soil K sequentially decreased its mobility as the fraction number increased.

Statistical analysis was performed using the SPSS v19.0 program (Chicago, U.S.). The soil K content of multiple fractions was analyzed using a two-way analysis of variance (ANOVA). Other indicators identified in the eight treatments were simply analyzed using one-way ANOVA. Differences between mean values were compared using the least significant difference (LSD) method at a significance threshold of 5%.

Simple nonlinear regression equations were also developed to evaluate the relationships among the response variables (K inputs and soil AK equilibrium). The experimental means were compared at the 95% probability level. Principal components analysis was performed using the Canoco 5.0 program for Windows to sort various indexes of soil K, including the K content of various fractions and the $M_F$ and $I_R$ values of soil K.

## Results

### Soil K content after a 60-d incubation

Soil AK, its sub-fractions (WSK, NSAK, and SAK), and NEK were all significantly influenced by K fertilization, straw addition, and a combination of the two (Table 1; $P<0.05$). After the 60-d incubation, K fertilization alone, single wheat straw addition, and single maize straw addition increased soil AK and NEK by 85.2% and 18.7%, 57.7% and 20.2%, and 41.5% and 15.5%, respectively, relative to the control. A combination of K fertilization and straw addition

**Table 1. ANOVA of the effects of K fertilization, straw return, and their interactions on soil K fractions after a 60-d incubation (P-values).**

| Source of variation | AK | | | | NEK | SK | TK |
|---|---|---|---|---|---|---|---|
| | WSK | NSAK | SAK | Total | | | |
| K fertilization (K1) | <0.001 | <0.001 | <0.001 | <0.001 | <0.001 | <0.001 | 0.048 |
| Straw addition (S) | <0.001 | <0.001 | <0.001 | <0.001 | <0.001 | <0.001 | ns |
| K1×S | <0.001 | <0.001 | <0.001 | <0.001 | 0.028 | ns | ns |

AK, available K; NEK, non-exchangeable K; SK, structural K; TK, total K; WSK, water-soluble K; NSAK, non-specific adsorptive K; and SAK, specific adsorptive K. ns indicates a non-significant difference ($P<0.05$).

**Table 2. Effects of K fertilization and straw addition on soil Available K (AK), Non-Exchangeable K (NEK), and Structural K (SK) after a 60-d incubation.**

| | AK (mg·kg⁻¹) | | NEK (mg·kg⁻¹) | | SK (g·kg⁻¹) | |
|---|---|---|---|---|---|---|
| | K0 | K1 | K0 | K1 | K0 | K1 |
| Control | 142 g | 263 d | 1254 d | 1488 bc | 18.4 A* | 17.9 AB |
| WS | 224 e | 383 b | 1507 b | 1531 b | 18.2 A | 17.9 AB |
| MS | 201 f | 286 c | 1448 c | 1550 ab | 18.3 A | 18.0 A |
| WS+MS | 266 d | 469 a | 1502 b | 1629 a | 17.5 B | 17.7 B |

Control, no straw addition; WS, wheat straw addition at 12 g straw·kg⁻¹ dry soil; MS, maize straw addition at 12 g straw·kg⁻¹ dry soil; WS+MS, wheat and maize straw addition, both at 12 g straw·kg⁻¹ dry soil. K0, no K fertilization; K1, K fertilization at 200 mg K₂O·kg⁻¹ dry soil. Significant differences are indicated by different case letters and * ($P<0.05$). Lower-case letters correspond to the interaction between K fertilization and straw addition; upper-case letters correspond to the effects of the four straw addition regimes; and * corresponds to the effect of the two K fertilization rates.

increased soil AK and NEK by 87.3–230% and 19.8–29.9%, respectively, relative to the control (Table 2; $P<0.05$). Moreover, soil AK was increased by interactions between K and WS and between K and WS+MS, whereas the interaction between K and MS decreased both the soil AK and NEK (Table 3; $P<0.05$). Additionally, soil SK was slightly decreased by K fertilization and straw addition (Table 2; $P<0.05$).

## Dynamics of soil K in available fractions

K fertilization increased soil WSK, NSAK, and SAK by 111.6%, 97.5%, and 42.7%, respectively, during the 60-d incubation (Fig 1). Similarly, straw addition also followed the same trend in increasing these three K fractions: WS+MS>WS>MS>Control.

**Table 3. Effects of the interaction between K fertilization and straw addition onsoil Available K (AK), Non-Exchangeable K (NEK), and Structural K (SK).**

| Type of interaction | AK (mg·kg⁻¹) | NEK (mg·kg⁻¹) | SK (g·kg⁻¹) |
|---|---|---|---|
| K1×WS | 19.0** | -105* | ns |
| K1×MS | -18.1** | -66* | ns |
| K1×(WS+MS) | 41.1** | ns | ns |

WS, wheat straw addition at 12 g straw·kg⁻¹ dry soil; MS, maize straw addition at 12 g straw·kg⁻¹ dry soil; WS+MS, wheat and maize straw addition, both at 12 g straw·kg⁻¹ dry soil. K1, K fertilization at 200 mg K₂O·kg⁻¹ dry soil.
** indicates an extremely significant difference ($P<0.01$).
* indicates a significant difference ($P<0.05$). ns indicates a non-significant difference ($P<0.05$).

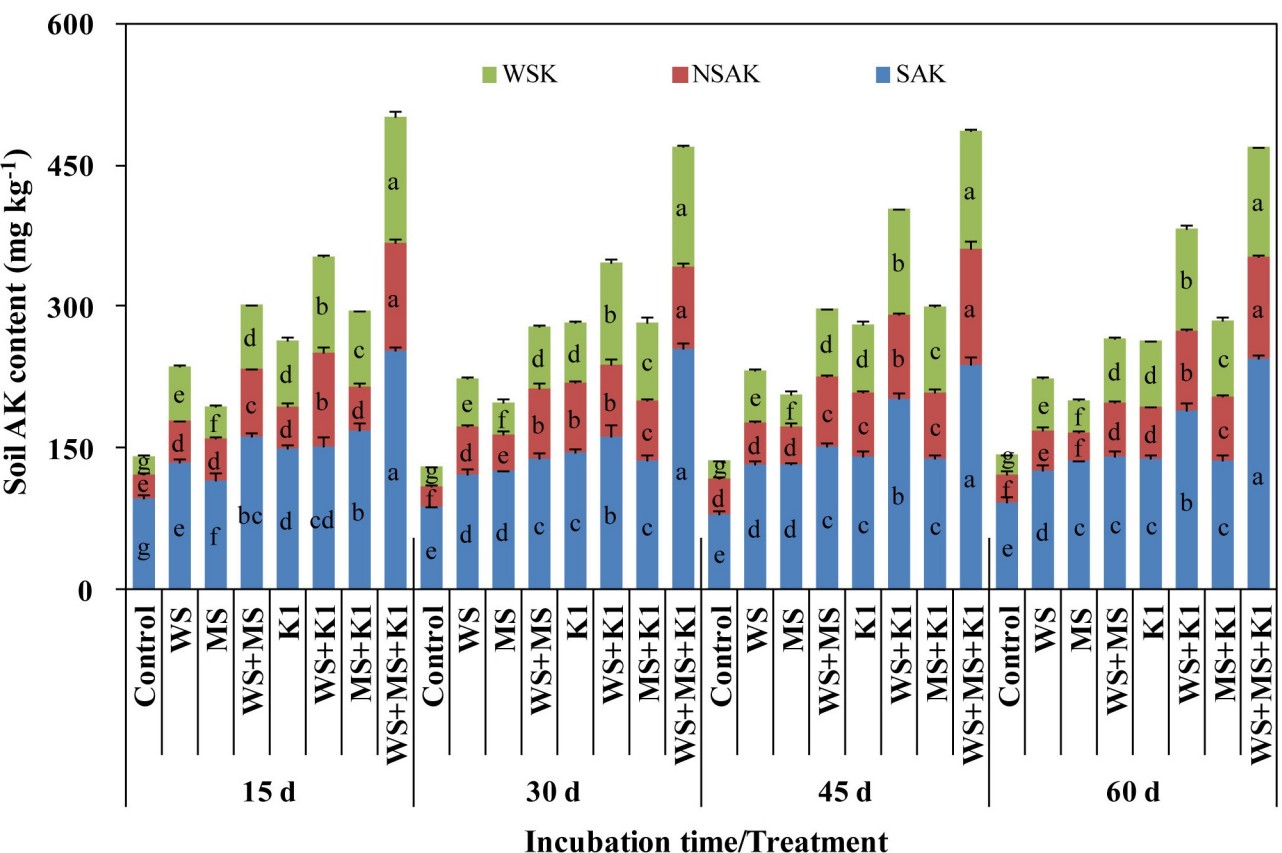

**Fig 1. Dynamics of soil K in available fractions during incubation following different treatments.** WSK, water-soluble K; NSAK, non-specific adsorptive K; SAK, specific adsorptive K. Control, no K addition; WS, wheat straw addition at 12 g straw·kg$^{-1}$ dry soil; MS, maize straw addition at 12 g straw·kg$^{-1}$ dry soil; WS+MS, wheat and maize straw addition, both at 12 g straw·kg$^{-1}$ dry soil; K1, K fertilization at 200 mg K$_2$O·kg$^{-1}$ dry soil. Different lower-case letters indicate significant differences among the eight treatments with respect to a certain K fraction during each incubation stage ($P<0.05$).

Soil AK content did not largely fluctuate during the 60-d incubation time (Fig 1). During the first few days (<15 d) after exogenous K addition, soil AK under each treatment condition remained relatively stable over time. Similar effects were seen with respect to soil WSK, NSAK, and SAK (Fig 1; $P<0.05$).

### Efficiency of exogenous K addition

The efficiency ratio of exogenous K addition was calculated to estimate the amount of K retained in the soil, which was shown to be 38.3–72.9% across the different treatments (Table 4). The highest efficiency ratio of exogenous K was derived from the mineral K fertilizer (72.9%), which was markedly higher than that derived from wheat (47.1%) or maize (39.3%) straw ($P<0.05$). Additionally, treatment with combined K sources also resulted in neutralized K availability ratios.

Regression analysis showed that the AK increased exponentially with increasing net K addition to soils with or without chemical K (Fig 2; $P<0.05$). Moreover, the rate of increase in soil AK was markedly higher following treatment with mineral K fertilization (K1) as compared with that with no mineral K fertilization (K0) (186.3>148.0 in regression coefficient).

**Table 4. Effects of K fertilization and straw addition on the efficiency ratio of exogenous K after a 60-d incubation.**

|  | Net K input (mg K·kg⁻¹ soil) | | Soil AK increase (mg·kg⁻¹) | | K efficiency ratio (%) | |
|---|---|---|---|---|---|---|
|  | K0 | K1 | K0 | K1 | K0 | K1 |
| Control | 0 | 166 | - | 121 d | - | 72.9 |
| WS | 174 | 340 | 82 e | 241 b | 47.1 | 70.9 |
| MS | 150 | 316 | 59 f | 144 c | 39.3 | 45.6 |
| WS+MS | 324 | 490 | 124 d | 327 a | 38.3 | 66.7 |

Control, no K addition; WS, wheat straw addition at 12 g straw·kg⁻¹ dry soil; MS, maize straw addition at 12 g straw·kg⁻¹ dry soil; WS+MS, wheat and maize straw addition, both at 12 g straw·kg⁻¹ dry soil. K0, no K fertilization; K1, K fertilization at 200 mg $K_2O$·kg⁻¹ dry soil. Lower-case letters indicate significant differences among the eight treatments ($P<0.05$).

## Changes in soil K mobility and stability

Soil SK made the largest contribution (89.4–93.0%) to total K as compared with the other fractions, whereas soil K in the WSK, NSAK, SAK, and NEK fractions centrally reflected the positive effects of exogenous K addition (Fig 3). In comparison with the control, the distributions of soil WSK, NSAK, SAK, and NEK, especially the K fractions with relatively higher mobility

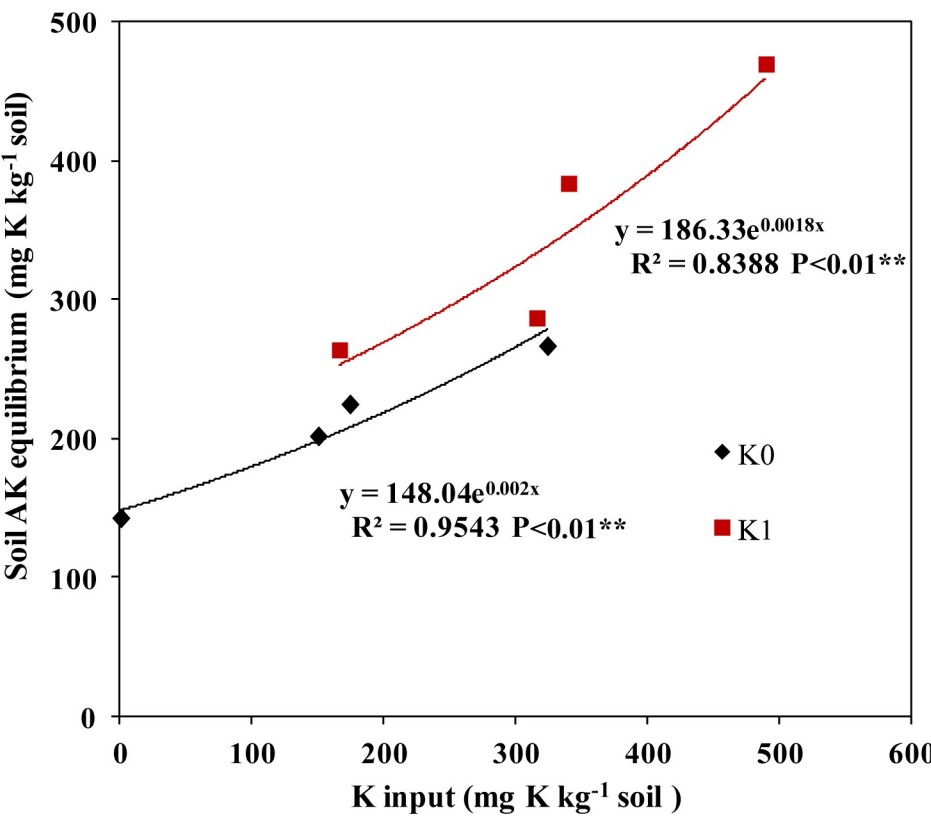

**Fig 2. Nonlinear regression performance of soil available K according to K addition.** K0, no K fertilization; K1, K fertilization at 200 mg $K_2O$·kg⁻¹ dry soil.

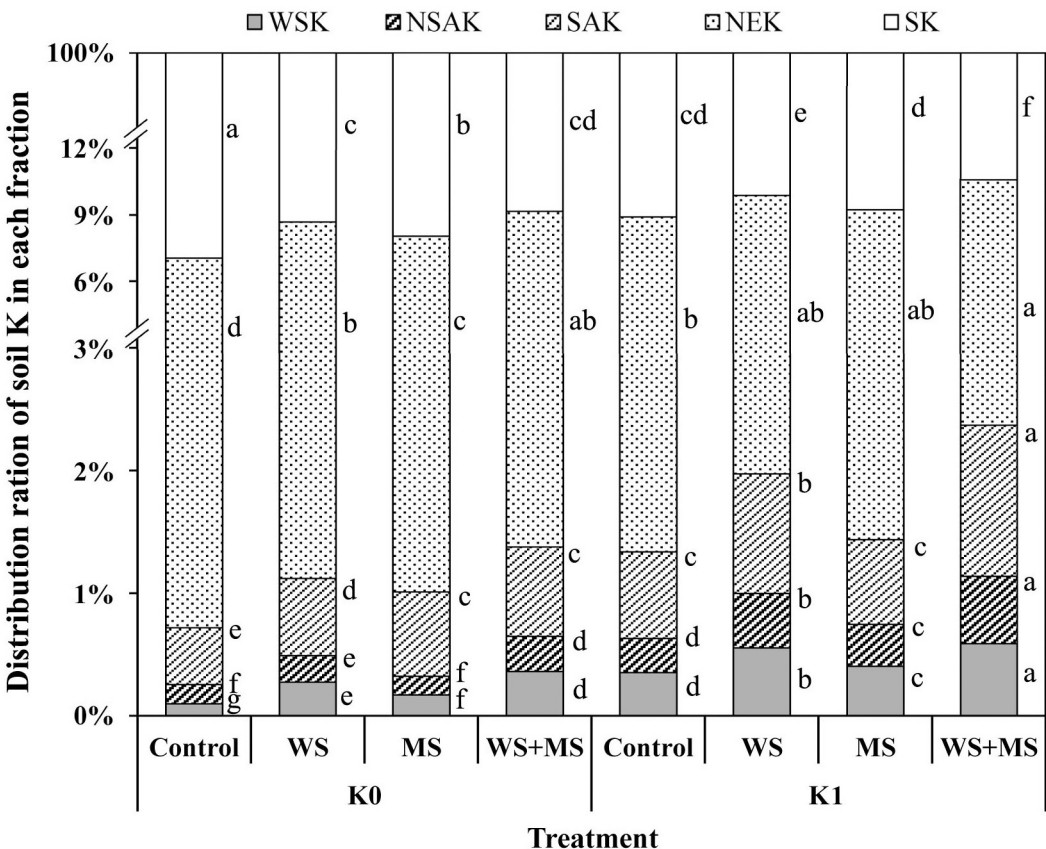

**Fig 3. Effects of exogenous K addition on the distribution ratio (%) of soil K in different factions: Water-Soluble K (WSK), Non-Specific Adsorptive K (NSAK), Specific Adsorptive K (SAK), Non-Exchangeable K (NEK), and Structural K (SK) following a 60-d incubation.** Control, no K addition; WS, wheat straw addition at 12 g straw·kg$^{-1}$ dry soil; MS, maize straw addition at 12 g straw·kg$^{-1}$ dry soil; WS+MS, wheat and maize straw addition, both at 12 g straw·kg$^{-1}$ dry soil; K1, K fertilization at 200 mg K$_2$O·kg$^{-1}$ dry soil. All eight treatments were used to determine the effects on soil K fraction distribution ratios using one-way ANOVA. Different lower-case letters indicate a significant difference among the eight treatments in each K fraction ($P<0.05$).

(readily available to plants), increased following mineral K fertilization, straw addition, and their combination, at the expense of soil SK ($P<0.05$).

Accordingly, the relative mobility factor ($M_F$) and reduced partition index ($I_R$) were calculated to comprehensively investigate the availability of soil K (Fig 4). In comparison with the control, both K fertilization and straw additions, as well as their combinations, increased the $M_F$ of soil K, whereas the $I_R$ value was decreased ($P<0.05$). Moreover, different straw addition regimes showed the following trend with respect to affecting both the $M_F$ and $I_R$ values of soil K: WS+MS>WS>MS>Control. Further, the highest $M_F$ and lowest $I_R$ of soil K were both observed following K fertilization (K1) among the three treatments with a single K source (K1, WS, and MS).

Principal components analysis (PCA) was performed to investigate the correlation between the $M_F$, $I_R$, and various fractions of soil K (Fig 5). The PC1 explained 94.31% of the composition variations in soil K across various exogenous K additions. The cluster containing all relatively available K fractions (AK, WSK, NSAK, SAK, and NEK) was well-loaded on the left axis of PC1 with the $M_F$ of soil K. In contrast, the other clusters containing relatively unavailable

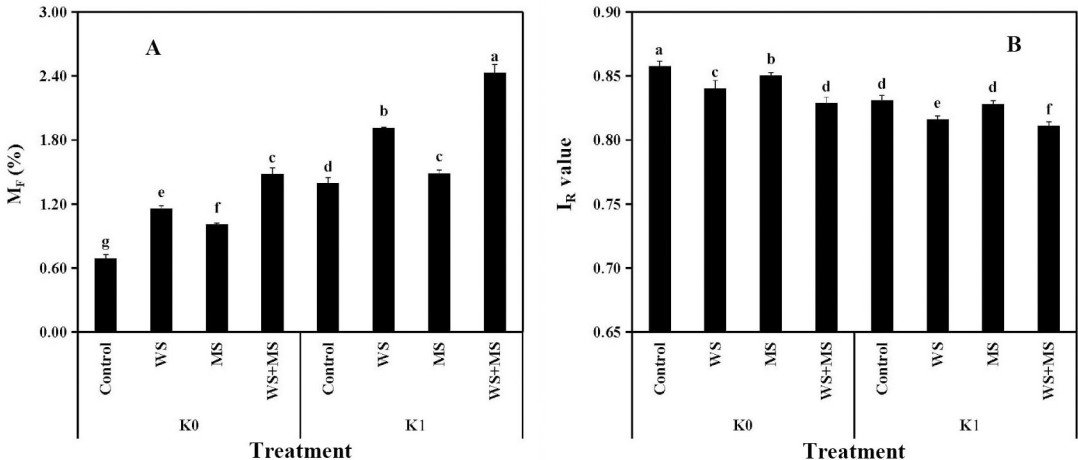

**Fig 4. Effects of K fertilization and straw addition on the mobility factor ($M_F$) (A) and reduced partition index ($I_R$) (B) for soil K during a 60-d incubation.** Abbreviations for K input strategies: Control, no K addition; WS, wheat straw addition at 12 g straw·kg$^{-1}$ dry soil; MS, maize straw addition at 12 g straw·kg$^{-1}$ dry soil; WS+MS, wheat and maize straw addition, both at 12 g straw·kg$^{-1}$ dry soil; K1, K fertilization at 200 mg K$_2$O·kg$^{-1}$ dry soil. Different lower-case letters indicate a significant difference among the eight treatments ($P$<0.05).

SK and TK were well-loaded on the right axis with the $I_R$ of soil K. The WSK, NSAK, SAK, and AK were all highly positively correlated with the $M_F$ of soil K, as was the NEK.

In addition, the treatments with a high net K input, especially WS+MS+K1 and WS+K1, were greatly responsible for the increases in both soil K in available fractions and $M_F$. On the contrary, the treatments with a relatively low net K (Control, WS, and MS) were responsible for an increase in the soil K stability and reserve (Fig 5).

## Discussion

### Effects of K addition on soil AK and NEK statuses

In comparison with K fertilization alone and single straw addition, a combination of K fertilization and straw addition resulted in larger increases in soil AK and NEK (Table 2 and Fig 3), implying that a greater net K addition can lead to greater increases in both soil AK and NEK. As mentioned previously, multiple factors including soil properties can affect soil AK levels and its fixation as NEK, with exogenous K addition and NEK release ultimately being the major sources of soil AK. In turn, fluctuating soil AK is also a primary factor that influences soil NEK through mineral fixation or release [29,33,46,47]. As is known, both mineral fertilizer K and crop straw K are released into the soil in a soluble form; therefore, irrespective of the K source, soil AK can be increased by increasing net K addition. As a result, soil AK was the most sensitive to K addition across the various fractions, responding more strongly to a combination of K fertilization and straw addition than to a single K source. Increased soil AK was also associated with increased exogenous K addition (Fig 2), which is consistent with previous studies [10,11,29]. However, our incubation experiment simulated practical K fertilization and straw return, and both were equivalent to almost triple the conventional application rates in field practices. The greatly increased soil AK cannot remain stable over time; instead, it is easily transformed into NEK with lower availability through soil K fixation [2,33,48]. With the exception of initial soil properties, soil K fixation capacity and rate are greatly associated with net K addition [2,29,46,49]. Soil K fixation capacity has been documented to increase with increasing net K addition, whereas the K fixation rate decreases [11,29]. That is, a relatively

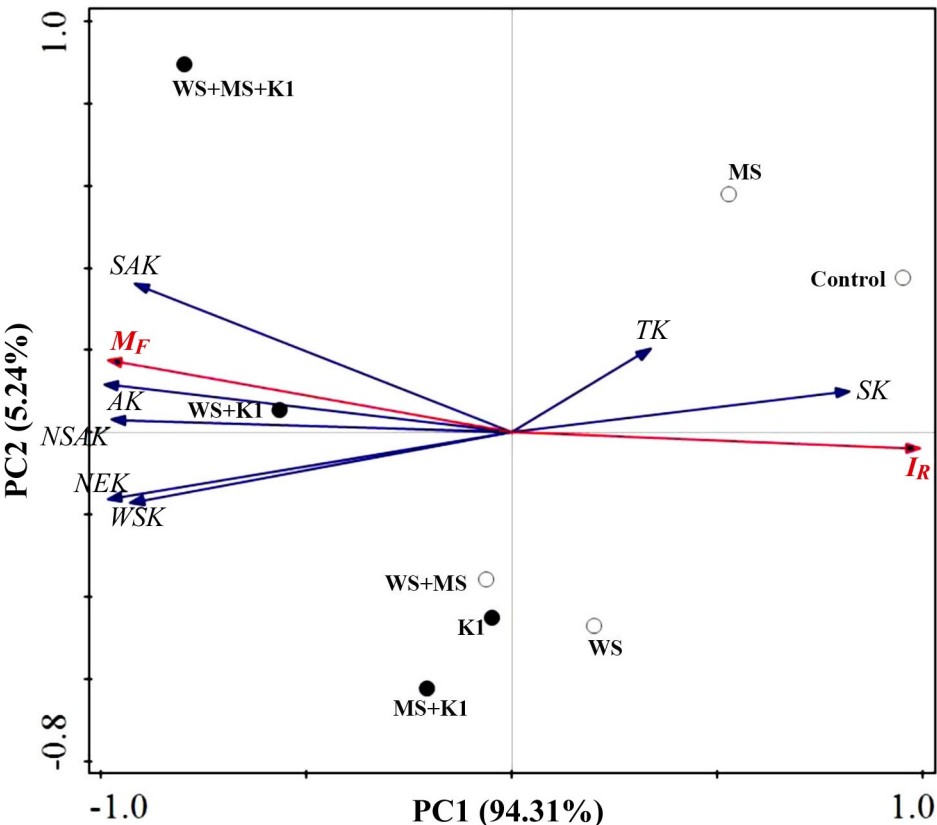

**Fig 5. Score scaling for principal components analysis, in which the indexes of soil K are represented by vectors and treatments are represented by symbols.** AK, available K; WSK, water-soluble K; NSAK, non-specific adsorptive K; SAK, specific adsorptive K; NEK, non-exchangeable K; SK, structural K; TK, total K. $M_F$ and $I_R$ represent the mobility factor and the reduced partition index for soil K, respectively. Control, no K addition; WS, wheat straw addition at 12 g straw·kg$^{-1}$ dry soil; MS, maize straw addition at 12 g straw·kg$^{-1}$ dry soil; WS+MS, wheat and maize straw addition, both at 12 g straw·kg$^{-1}$ dry soil; K1, K fertilization at 200 mg K$_2$O·kg$^{-1}$ dry soil.

higher K addition easily results in a larger soil K fixation capacity but retains a higher efficiency in increasing soil AK [46,49,50]. In the present study, a combination of K fertilization and straw addition with relatively higher K inputs generally resulted in larger increases in soil NEK, which closely responded to fluctuating soil AK (Table 2).

Concomitantly, lower efficiencies in increasing soil AK were observed after crop straw (especially maize straw) addition as compared with those after K fertilization (Table 4), which may be partially due to the lower amount of K derived from crop straw (especially maize straw) than from K fertilizer. However, relatively smaller differences in net K addition among the three K sources (K fertilizer, wheat straw, and maize straw) led to relatively larger differences in efficiency in increasing soil AK (Table 4). This indicates that, with the exception of K fixation in soil minerals, soil AK reduction can be caused by other factors, such as changes in the physical and biological properties of soil, which require further exploration [29,51,52].

In contrast, soil SK decreased slightly after exogenous K addition, especially following treatment with a high amount of K (Table 2). Similar results have rarely been reported. We speculate that the incubation conditions in the present study may be the main factor driving SK release due to its relatively high soil moisture and stable temperature, which are beneficial to

soil mineral weathering without crop growing. Thus, in addition to the external factors influencing soil K availability, further research is required to reveal the internal mechanisms of soil K transformation among different fractions.

### Effects of crop straw decomposition on soil AK status

Straw addition was not as efficient as K fertilization in increasing the available K level in the soil (Table 3), which has not been well documented in previous studies. Typically, K exists in an ionic form in plants and can be easily released from decaying crop residue, becoming available for subsequent crops [21]; however, crop residue decomposition is certainly an important biochemical process dominated by soil microorganisms. Recent research has reported that the addition of organic material can trigger microbial activity and extracellular enzyme production, causing the decomposition of both newly added organic material and certain fractions of native SOC [53]. The increased number of microorganisms absorb soil available nutrients for metabolism and form microbial byproducts, implying reduced soil available nutrients, including extractable metals, with regard to microbial immobilization [37,51,52,54,55]. Moreover, the increased active SOC, with a lower degree of decomposition and a rich source of polar functional groups, can drive the formation of organo-mineral complexes, thus improving soil aggregate stability [38,56,57]. Soil aggregation can also help to protect soil available nutrients through physical occlusion [38,52,55]. Therefore, we speculated that although the increased SOC may be associated with increased adsorptive sites to protect AK against fixation, soil AK may also be reduced by microbial immobilization and aggregate occlusion. Maybe that's why straw addition had a markedly lower efficiency in increasing soil AK than K fertilization in the present study (Table 3).

In addition, wheat straw differed from maize straw in efficiency in increasing soil AK (Table 3), which may be attributed to the different chemical compositions of the two types of straw. In the present research, the relatively higher efficiency of wheat straw in increasing soil AK may result from its higher C/N ratio (80:1) and higher lignin content (20%) than that of maize straw (57:1 C/N ratio and 14% lignin content). According to the microbial stoichiometry and metabolism theory, microbial activity is higher when the low C/N ratio (or high N availability) of substrates matches the microbial demand or when the compounds are easily decomposed by microorganisms, increasing both the enzyme production and organic C degradation [58–61]. Therefore, the relatively appropriate composition of maize straw may alleviate the metabolic constraint in the soil and improve the microbial activity and soil quality, but which can cause available K immobilization, leading to a lower efficiency in increasing soil AK than that of wheat straw. In addition, the results showed that the increases in soil AK and its sub-fractions (WSK, NSAK, and SAK) were stable during the first 15 days following straw addition and K fertilization (Fig 1). This suggests that the increased SOC may not be sufficient to overcome the multiple AK reduction forces, including microbial utilization, aggregate occlusion, and mineral fixation [22,29,31]. Moreover, the results also indicate that crop straw K can be easily released from decaying straw due to the commonly ionic form of K being present in plants. Straw decomposition did not slow the straw K release rate, whereas it greatly caused AK reduction through microbial and aggregation pathways. However, further studies regarding the relationship between soil K dynamics and straw decomposition, in addition to soil microbial activity, are required to verify our inferences.

### Soil K redistribution and bioavailability

Soil metal elements present in each fraction dictate their specific mobility. The content of the extractable and exchangeable fractions of a metal may be described by the mobility factor ($M_F$)

[28,36,41,42,45]. Historically, WSK and EK (NSAK+SAK) with high mobility are also considered available to plants [1,2,31]. In agricultural practice, increased crop production is associated with increased soil AK (WSA+EK) [10,11,23,24]. Our results show that soil AK and its sub-fractions (WSK, NSAK, and SAK) increased following increases in exogenous K addition (Table 2 and Figs 1–3), thereby driving an increase in the soil K $M_F$ value (Fig 4). Although no crop was planted in the present incubation in response to the increased soil AK level, the soil K bioavailability was undoubtedly increased [41,42,45].

Nevertheless, the simultaneously increased soil NEK with lower mobility (relative to AK) should be equally considered when assessing soil K availability in long-term cropping (Table 2). As mentioned previously, metal transformation in all individual fractions concerns redistribution and can be expressed by the reduced partition index ($I_R$), which was introduced to quantitatively describe the relative binding intensities of soil metals. In contrast to $M_F$, $I_R$ not only includes metal transformation among labile fractions but also among those that are stable [41,42,45]. Since the binding intensity of soil K, in turn, decreases from the F1 to F5 fractions, it is reasonable to suggest that this index ($I_R$) is appropriate for assessing soil K redistribution. The calculated $I_R$ of soil K ranged from 0.04 to 1; a high value indicates soil K stability resulting from its occurrence in non-exchangeable (F4, NEK) and structural (F5, SK) fractions, and a low value represents a distribution pattern with a high proportion of exchangeable (F2 and F3, NSAK and SAK) and soluble (F1, WSK) fractions. In the present study, the $I_R$ of soil K was reliably decreased following increases in K addition, implying that increasing K input successfully increased the soil K bioavailability (Fig 4).

Moreover, the soil K $M_F$ value was positively correlated with the K in all relatively available fractions (AK and its sub-fractions, as well as NEK), whereas the soil K $I_R$ value was negatively correlated with all relatively available fractions of soil K (Fig 5). This implies that the use of both the $M_F$ and $I_R$ values of soil K is feasible to quantitatively assess soil K mobility, and thus determine soil K bioavailability [10,31,41,42]. The present results indicate that crop straw addition, specifically maize straw, was less efficient in increasing soil K bioavailability than K fertilization. However, the soil K bioavailability increased the most following the highest net K addition, which was a combination of straw addition and K fertilization (Fig 5). Therefore, when considering the K uptake by crops in agricultural practice, specifically in intensive agricultural systems in China with commonly practiced straw return [25,26], straw return should be combined with K fertilization to improve both soil K bioavailability and K cycling in soil–plant systems.

## Conclusions

The present study suggests that K fertilization and crop straw addition were efficient in rapidly increasing soil K in fractions available to plants. Different K sources showed the following trend in increasing soil K availability: K fertilizer>wheat straw>maize straw. However, soil AK generally increased with an increase in net K addition, and the three K sources resulted in larger increases in both soil AK and NEK when used in combination as compared with being applied individually. Positive correlations existed between the $M_F$ value and each relatively available fraction of soil K, including WSK, NSAK, SAK, and NEK; and negative correlations existed between the $I_R$ value and each relatively available fraction of soil K. Therefore, the $M_F$ and $I_R$ values of soil K can be used to assess the comprehensive availability of soil K. K fertilization in combination with crop straw return appears to be the optimal method for improving soil K availability and K cycling in soil–plant systems.

## Supporting information

**S1 Dataset.**
(ZIP)

## Author Contributions

**Conceptualization:** Xiushuang Li, Jianglan Shi, Xiaohong Tian.

**Data curation:** Xiushuang Li.

**Formal analysis:** Xiushuang Li, Yafei Li, Tianqi Wu, Chunyan Qu.

**Funding acquisition:** Xiushuang Li, Jianglan Shi, Xiaohong Tian.

**Investigation:** Xiushuang Li, Peng Ning, Jianglan Shi, Xiaohong Tian.

**Methodology:** Xiushuang Li, Yafei Li, Tianqi Wu, Chunyan Qu.

**Project administration:** Xiushuang Li, Jianglan Shi, Xiaohong Tian.

**Resources:** Xiushuang Li, Yafei Li, Tianqi Wu, Chunyan Qu, Peng Ning.

**Software:** Xiushuang Li, Yafei Li, Tianqi Wu, Chunyan Qu.

**Supervision:** Xiushuang Li, Jianglan Shi.

**Validation:** Xiushuang Li.

**Visualization:** Xiushuang Li.

**Writing – original draft:** Xiushuang Li, Jianglan Shi, Xiaohong Tian.

**Writing – review & editing:** Xiushuang Li, Peng Ning, Jianglan Shi, Xiaohong Tian.

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
