## [Decision Letter · Decision Letter 0]

25 May 2020

PONE-D-20-11140

Potassium fertilization combined with crop straw incorporation changes soil potassium fractions and availability in northwest China

PLOS ONE

Dear Dr. Li,

Thank you for submitting your manuscript to PLOS ONE. After careful consideration, we feel that it has merit but does not fully meet PLOS ONE’s publication criteria as it currently stands. Therefore, we invite you to submit a revised version of the manuscript that addresses the points raised during the review process.

We look forward to receiving your revised manuscript.

Kind regards,

Vassilis G. Aschonitis

Academic Editor

PLOS ONE

Journal Requirements:

Reviewers' comments:

Reviewer's Responses to Questions

**Comments to the Author**

1. Is the manuscript technically sound, and do the data support the conclusions?

Reviewer #1: Partly

2. Has the statistical analysis been performed appropriately and rigorously? 

Reviewer #1: Yes

3. Have the authors made all data underlying the findings in their manuscript fully available?

Reviewer #1: Yes

4. Is the manuscript presented in an intelligible fashion and written in standard English?

Reviewer #1: Yes

5. Review Comments to the Author

Reviewer #1: The MS PONE-D-20-11140 "Potassium fertilization combined with crop straw incorporation changes soil potassium fractions and availability in northwest China" addresses, using an incubation study, the changes in K fractions after crop residues (corn or wheat or combined ) application with K fertilization. The work not novel but it addresses a specific situation in China. The MS still needs much improvements.

Here are general comments but for specific ones please refer to the attached file. Given there were 2 files submitted in the PDF the reviews and Ln here refers to that of the New Manuscript Word file (Page 36 of the PDF)

Ln 2 Title: should be changed, see the attached file

Ln 22 change chemical K to mineral K, here and elsewhere

Ln 35 Decipher Mf and IR

Ln 138 Can you provide an average of K content (available and/or total)

Ln 152 Use another symbol (e.g., K1). reader would get confused whether K refers to 200 mg K2O or to potassium

Ln 154 What the content of K in these organic substrate.. These materials will decompose differently (Wheat C:N is 80:1 while maize C/N is 60:1 as per the USDA)

Ln 426-35 SAK was always higher after K+WS than after K+MS which may contradict the statement below. Either wheat had a higher content of K (selective absorption of K by wheat) or the urea that was added accelerated the decomposition of wheat straw given it is C rich.

6. PLOS authors have the option to publish the peer review history of their article (what does this mean?). If published, this will include your full peer review and any attached files.

Reviewer #1: Yes: Hamada Abdelrahman

---

## [Author Response · Author response to Decision Letter 0]

4 Jun 2020

Dear Reviewer(s),

Thank you for your constructive and positive comments, which were valuable and helpful for revising and improving the quality of our paper. After carefully considering your comments, we revised our manuscript accordingly. We would like submit a revised version of the manuscript that addresses the points raised during the review process, and hope our revisions meet or exceed the expectations of the Editor(s) and Reviewers.

Our point-by-point responses to the Editor(s) and Reviewers’ comments are as follows:

1 Ln 2 Title: should be changed, see the attached file.

Response: Thank you very much for your valuable comments. We have re-phrased the title accordingly. We improved the title of article according to your comments by changing the "changes" to "improves". We think that this can well generalize our research and conclusions.

2 Ln 22 change chemical K to mineral K, here and elsewhere.

Response: We have changed the "chemical K" to "mineral K" throughout our manuscript. 

3 Ln 35 Decipher Mf and IR.

Response: Thank you very much for your review of our manuscript and your valuable suggestion. We apologize for this omission. We have revised this section according to your comment, deciphered both the MF and IR at their first appearances.

4 Ln 138 Can you provide an average of K content (available and/or total).

Response: We have added the total K data according to your suggestion. Please see in Line 139.

5 Ln 152 Use another symbol (e.g., K1). reader would get confused whether K refers to 200 mg K2O or to potassium.

Response: We thanks for your valuable suggestion and apologize for this omission. We have changed the abbreviation of K fertilization from "K" to "K1" throughout our manuscript.

6 Ln 154 What the content of K in these organic substrate. These materials will decompose differently (Wheat C:N is 80:1 while maize C/N is 60:1 as per the USDA).

Response: We have provided the K concentration in wheat and maize straws according to your comments. Please see the section in Line 149-150.

7 Ln 426-435 SAK was always higher after K+WS than after K+MS which may contradict the statement below. Either wheat had a higher content of K (selective absorption of K by wheat) or the urea that was added accelerated the decomposition of wheat straw given it is C rich.

Response: We apologize for this lack of clarity and have improved this section according to your suggestions. The original intention of this paragraph was that compared with wheat straw (80:1 C/N ratio and 20% lignin content), maize straw (57:1 C/N ratio and 14% lignin content) is easier to decompose and promote the activity of microorganisms in the soil, thus improving the bio-chemical properties of soil. However, increased soil quality may promote the immobilization of soil available nutrients through multiple factors, including microbial utilization, aggregate occlusion, etc. The SAK is one of the sub-fractions of soil available K, and which is closely associated with soil AK. Thus, the SAK was always higher after K+WS than after K+MS.

In our research, the tested wheat straw (14.5% K content) did not have a much higher content of K than maize straw (12.5% K content), and the straws both had a total C content of about 40%. In addition, both the added urea and superphosphate to the incubated soil were equal among all treatments, which supplied available nutrients for microbial metabolism. For these, we did not think that the slight difference of K content in added straws and the equal urea input in all treatments can result in a large difference of K efficiency ratio existing between wheat and maize straw addition. In contrast, the difference in quality between wheat and maize straws seems to explain our results here.

I have good faith that the quality of the manuscript is further improved and will be ready for publication in PLOS ONE. Please let me know if you have any questions.

Best regards,

Xiushuang Li

---

## [Decision Letter · Decision Letter 1]

1 Jul 2020

PONE-D-20-11140R1

Potassium fertilization combined with crop straw incorporation improves soil potassium fractions and availability in northwest China

PLOS ONE

Dear Dr. Li,

Thank you for submitting your manuscript to PLOS ONE. After careful consideration, we feel that it has merit but does not fully meet PLOS ONE’s publication criteria as it currently stands. Therefore, we invite you to submit a revised version of the manuscript that addresses the points raised during the review process.

We look forward to receiving your revised manuscript.

Kind regards,

Vassilis G. Aschonitis

Academic Editor

PLOS ONE

Reviewers' comments:

Reviewer's Responses to Questions

**Comments to the Author**

1. If the authors have adequately addressed your comments raised in a previous round of review and you feel that this manuscript is now acceptable for publication, you may indicate that here to bypass the “Comments to the Author” section, enter your conflict of interest statement in the “Confidential to Editor” section, and submit your "Accept" recommendation.

Reviewer #1: (No Response)

2. Is the manuscript technically sound, and do the data support the conclusions?

Reviewer #1: Partly

3. Has the statistical analysis been performed appropriately and rigorously? 

Reviewer #1: Yes

4. Have the authors made all data underlying the findings in their manuscript fully available?

Reviewer #1: Yes

5. Is the manuscript presented in an intelligible fashion and written in standard English?

Reviewer #1: (No Response)

6. Review Comments to the Author

Reviewer #1: The current draft addressed most of the comments on the previous submission, however, authors still need to address the following:

i) the results, conclusion are mainly based on 60 d lab incubation of a 250 g soil . That is not sufficient for drawing a conclusion on decomposition of organic substrates and altering K fractions, at least for many of top expert in the field. I suggested that the title include "an incubation study” but authors did not address it. Also, the word improve in the titles does not really reflect what really happened. What does it mean to improve "soil non-specific adsorptive K" or "NEK"?does it mean to increase it or decrease it.

ii) the comment on section starting Ln 417 on SAK was always higher after K+WS than after K+M have speculations on immobilization and microbial activity that are stated as a fact. As these were not lab or field verified, it should be stated as speculation

7. PLOS authors have the option to publish the peer review history of their article (what does this mean?). If published, this will include your full peer review and any attached files.

Reviewer #1: No

---

## [Author Response · Author response to Decision Letter 1]

3 Jul 2020

Dear Reviewer(s),

Thanks very much for your constructive and positive comments, which were valuable and helpful for revising and improving the quality of our paper. After carefully considering your comments, we revised our manuscript accordingly. We would like submit a revised version of the manuscript that addresses the points raised during the review process, and hope our revisions meet or exceed the expectations of the Editor(s) and Reviewers.

Our point-by-point responses to the Editor(s) and Reviewers’ comments are as follows:

i) the results, conclusion are mainly based on 60 d lab incubation of a 250 g soil . That is not sufficient for drawing a conclusion on decomposition of organic substrates and altering K fractions, at least for many of top expert in the field. I suggested that the title include "an incubation study” but authors did not address it. Also, the word improve in the titles does not really reflect what really happened. What does it mean to improve "soil non-specific adsorptive K" or "NEK"?does it mean to increase it or decrease it.

Response: We apologize for this lack of clarity and have improved this section according to your suggestions. We originally thought that although the present study was conducted using an incubation experiment, there was no need to name methods in the title, instead, it would make the title lengthy. However, after reading your comments carefully this time, we think it is really necessary to name the research method in the title in view of the limitations of this study. Thank you very much for your valuable comments. We have re-phrased the title accordingly, and changed the "improves" to "alters". We think that this can well generalize our research and conclusions.

ii) the comment on section starting Ln 417 on SAK was always higher after K+WS than after K+MS have speculations on immobilization and microbial activity that are stated as a fact. As these were not lab or field verified, it should be stated as speculation.

Response: Thank you very much for your review of our manuscript and your valuable suggestion. We apologize for this omission. We also think that it is necessary to indicate that it is a speculation when analyzing theories that not directly related to the present study, so as to ensure the accuracy and rigor of the study. We have revised this section according to your comment.

I have good faith that the quality of the manuscript is further improved and will be ready for publication in PLOS ONE. Please let me know if you have any questions.

---

## [Decision Letter · Decision Letter 2]

13 Jul 2020

Potassium fertilization combined with crop straw incorporation alters soil potassium fractions and availability in northwest China: An incubation study

PONE-D-20-11140R2

Dear Dr. Li,

We’re pleased to inform you that your manuscript has been judged scientifically suitable for publication and will be formally accepted for publication once it meets all outstanding technical requirements.

Kind regards,

Vassilis G. Aschonitis

Academic Editor

PLOS ONE

Additional Editor Comments (optional):

Reviewers' comments:

Reviewer's Responses to Questions

**Comments to the Author**

1. If the authors have adequately addressed your comments raised in a previous round of review and you feel that this manuscript is now acceptable for publication, you may indicate that here to bypass the “Comments to the Author” section, enter your conflict of interest statement in the “Confidential to Editor” section, and submit your "Accept" recommendation.

Reviewer #1: All comments have been addressed

2. Is the manuscript technically sound, and do the data support the conclusions?

Reviewer #1: Yes

3. Has the statistical analysis been performed appropriately and rigorously? 

Reviewer #1: Yes

4. Have the authors made all data underlying the findings in their manuscript fully available?

Reviewer #1: Yes

5. Is the manuscript presented in an intelligible fashion and written in standard English?

Reviewer #1: Yes

6. Review Comments to the Author

Reviewer #1: (No Response)

7. PLOS authors have the option to publish the peer review history of their article (what does this mean?). If published, this will include your full peer review and any attached files.

Reviewer #1: **Yes: **Hamada Abdelrahman

---

## [Editor Report · Acceptance letter]

15 Jul 2020

PONE-D-20-11140R2 

Potassium fertilization combined with crop straw incorporation alters soil potassium fractions and availability in northwest China: An incubation study 

Dear Dr. Li:

I'm pleased to inform you that your manuscript has been deemed suitable for publication in PLOS ONE. Congratulations! Your manuscript is now with our production department. 

Kind regards, 

on behalf of

Dr. Vassilis G. Aschonitis 

Academic Editor

PLOS ONE